# COPD, Pulmonary Fibrosis and ILAs in Aging Smokers: The Paradox of Striking Different Responses to the Major Risk Factors

**DOI:** 10.3390/ijms22179292

**Published:** 2021-08-27

**Authors:** Bianca Beghé, Stefania Cerri, Leonardo M. Fabbri, Alessandro Marchioni

**Affiliations:** 1Respiratory Diseases Unit, University Hospital of Modena, 41124 Modena, Italy; stefania.cerri@unimore.it (S.C.); alessandro.marchioni@unimore.it (A.M.); 2Department of Translational Medicine and Romagna, University of Ferrara, 44121 Ferrara, Italy; leonardo.fabbri20@gmail.com

**Keywords:** dyspnea, emphysema, inflammation, genetic, exacerbations, non-invasive mechanical ventilation

## Abstract

Aging and smoking are associated with the progressive development of three main pulmonary diseases: chronic obstructive pulmonary disease (COPD), interstitial lung abnormalities (ILAs), and idiopathic pulmonary fibrosis (IPF). All three manifest mainly after the age of 60 years, but with different natural histories and prevalence: COPD prevalence increases with age to >40%, ILA prevalence is 8%, and IPF, a rare disease, is 0.0005–0.002%. While COPD and ILAs may be associated with gradual progression and mortality, the natural history of IPF remains obscure, with a worse prognosis and life expectancy of 2–5 years from diagnosis. Acute exacerbations are significant events in both COPD and IPF, with a much worse prognosis in IPF. This perspective discusses the paradox of the striking pathological and pathophysiologic responses on the background of the same main risk factors, aging and smoking, suggesting two distinct pathophysiologic processes for COPD and ILAs on one side and IPF on the other side. Pathologically, COPD is characterized by small airways fibrosis and remodeling, with the destruction of the lung parenchyma. By contrast, IPF almost exclusively affects the lung parenchyma and interstitium. ILAs are a heterogenous group of diseases, a minority of which present with the alveolar and interstitial abnormalities of interstitial lung disease.

## 1. Introduction

Aging and smoking are associated with the development of three main pulmonary diseases: chronic obstructive pulmonary disease (COPD); interstitial lung abnormalities (ILAs), including combined pulmonary fibrosis and emphysema (CPFE); and idiopathic pulmonary fibrosis (IPF). All manifest almost exclusively after the age of 60 years, with very different prevalence: COPD up to 40% [1], ILAs 8–9% [2,3], CPFE < 1%, and IPF, a rare disease, 0.0005–0.002% [4], although this may rise sharply after the age of 70 years [5]. Different from COPD, CPFE, and IPF, only a minority of non-CPFE ILAs are progressive, with these variants representing a form of early interstitial lung disease (ILD), and it is therefore important from the outset to distinguish progressive from nonprogressive ILAs [6].

The focus of this perspective is to discuss the paradox of the striking pathological and pathophysiologic responses on the background of the two main risk factors, i.e., aging and smoking. COPD is characterized by small airways fibrosis and remodeling, with the destruction of the lung parenchyma, whereas IPF almost exclusively affects the lung parenchyma and interstitium. We will also discuss some aspects of CPFE.

## 2. Chronic Obstructive Pulmonary Disease

Aging in non-smokers is associated with pathological and physiological changes that closely resemble COPD, with a decrease in forced expiratory volume in 1 s (FEV_1_) more than in forced vital capacity (FVC), and consequently, a decrease in FEV_1_/FVC ratio, accompanied by an increase in residual volume (RV) at the expense of FVC, and thus with no change in total lung capacity [7]. Interestingly, combined aging and smoking are associated with slow but exponentially increasing airflow limitation over time, which is perceived by the patients almost exclusively after the age of 40 years but is usually diagnosed after 60 years of age. In about half of the patients with COPD, the development of airflow limitation occurs due to an excessive decline in FEV_1_ after the age of 25, whereas in the remainder, it occurs without an excess decline in individuals who already have reduced lung function in early adulthood [8] due to early events [9,10]. Independently from the heterogeneous decline in lung function, the natural history of clinical COPD in the general population has been recently described, with an incidence of roughly 0.5, 1.0, and 1.6% in younger (35–54 years), middle (55–64 years), and older (>65 years) adults, respectively, with prevalence in the same study of approximately 5, 11, and 23%, respectively [11] (Figure 1).

While very interesting, as they utilize a robust definition of ’clinical COPD’, these percentages underestimate the natural history of the disease, as the definition used has a low sensitivity (56%) and values are calculated in the general population, i.e., independently from smoking and other risk factors. However, the results are informative, as they suggest a progressive increase in COPD prevalence that fits very well with the estimate of 41% of spirometrically confirmed COPD (Global Initiative for Chronic Obstructive Lung Disease [GOLD] stage 2 [12]*,* i.e., FEV_1_/FVC <0.7 and FEV_1_ <80% predicted) reported by the COPDGene study that included only middle-aged and older (45–80 years) adult smokers (with a smoking history of >10 packs/year) [1].

The three main characteristics of patients with COPD typically identified in the clinic are older age, smoking, and male sex [12]. The main clinical manifestation is progressive dyspnea, associated with productive (chronic bronchitis) or dry sputum in 20–40% of individuals, and recurrent exacerbations of the respiratory symptoms in up to 30% that are important causes of morbidity and mortality [12,13].

The development of COPD involves structural abnormalities of the lung parenchyma and airways, with the destruction of the alveolar walls (emphysema) and destruction and remodeling of the peripheral airways (bronchiolitis) [14]. These are induced by an abnormal chronic respiratory reaction to smoking amplified by inflammaging, the immunological changes that occur with aging [15,16]. Concomitant chronic diseases (e.g., cardiovascular, metabolic, neurological, etc.) are frequent due to the shared risk factors [17].

Although aging and smoking are mainly associated with the development of obstructive abnormalities of the lung, in up to 10% of individuals, the functional abnormality may be combined with the development of interstitial lung abnormalities [1] that either mask the obstructive abnormality without a significant reduction in lung volumes, or are associated with a clear restrictive pattern [1,6].

In conclusion, COPD is the most frequent chronic pulmonary disease in elderly smokers. It is characterized by inflammation and destruction of airways and lung parenchyma, causing a pathognomonic, poorly reversible airflow limitation (obstructive spirometric pattern) clearly distinguishable for IPF and other ILDs that are characterized by a reduction in lung volumes (restrictive spirometric pattern). ILAs show a heterogeneous pattern.

## 3. Interstitial Lung Abnormalities and Combined Pulmonary Fibrosis and Emphysema

ILAs are areas of increased lung density on computed tomography (CT) scans that, similar to COPD, can result from tobacco smoking. [1,3]. A minority of ILAs may be considered a potentially progressive ILD that may evolve in clinically overt ILD when they are characterized by chest CT evidence of honeycombing or traction bronchiectasis/bronchiolectasis [6]. The characteristics of smokers with and without ILA have been described in a pivotal study by Washko and Colleagues [1]. According to this study, 19% of patients with ILA have centrilobular abnormalities, 55% subpleural abnormalities, 20% combined centrilobular and subpleural abnormalities, and only 6% have radiographic interstitial lung disease. Smokers with ILAs have a reduced total lung capacity (the extent of which varies according to ILA subtype), with a minority at an increased risk of a restrictive ventilatory defect. However, not surprisingly, considering the nature of the structural abnormalities of the lungs, smokers with ILAs and COPD may be at increased risk of death than subjects without ILA [3]. In conclusion, ILAs are complex airway/parenchymal/interstitial structural abnormalities, not yet clearly clinically or pathologically defined. Expert panels are developing recommendations on the identification, referral, and follow-up of ILAs [6].

CPFE is usually not included in the ILAs; we propose its inclusion here to avoid a separate section. CPFE is a syndrome occurring in smokers or ex-smokers characterized by dyspnea that is often severe, with relatively preserved lung volumes, severely impaired gas exchange, and an increased risk of pulmonary hypertension, and is associated with a very poor prognosis, including an increased risk of lung cancer [18,19,20,21]. It is also encountered in the setting of connective tissue diseases, especially rheumatoid arthritis, which has similar features to ‘idiopathic’ (tobacco-related) CPFE [18]. The diagnosis is based on the presence of emphysema predominating in the upper lobes (frequently paraseptal) together with interstitial abnormalities, suggesting pulmonary fibrosis in the lower lung zones, with velcro crackles on auscultation. The combination of emphysema and fibrosis may result in relatively normal spirometry (FEV_1_ and FVC), with a significantly reduced gas diffusion co-efficient. CPFE may be associated with a higher risk of pulmonary hypertension and death (although this is controversial) [22]. Despite the publication of several papers since the first description by Wiggins et al. [21], the pathophysiology and mechanisms of CPFE largely remain obscure [18,20].

## 4. Idiopathic Pulmonary Fibrosis

IPF is a chronic and progressive fibrosing interstitial pneumonia of unknown etiology occurring primarily in older adults, with thoracic CT scans characteristically showing usual interstitial pneumonia (UIP). IPF is driven by aberrantly activated epithelial cells [23,24,25], with a histopathological pattern characterized by temporal and spatial heterogeneity and areas of normal-appearing lung parenchyma in close proximity to fibrotic areas with honeycombing, resulting in a marked increase in lung stiffness, scarring, and decreased lung volume [23,24,25]. Similar to COPD, IPF is associated with concomitant chronic diseases [26] and exacerbations, the latter with very different characteristics and prognoses [27].

IPF accounts for 20–50% of all cases of ILD and represents the most frequent and severe of idiopathic interstitial pneumonia (IIPs), a group of ILDs of unknown cause [28]. IPF occurs in less than 5 per 10,000 persons/year, with a prevalence ranging between 0.0005 and 0.002% [4], so tiny compared to COPD (up to 40%) or ILAs (up to 9%) [1]. However, its burden is high, with a very poor prognosis and median survival of 3–5 years [4]. Thus, despite the number of patients with IPF being small, the extremely high individual, social, and direct and indirect costs mean that IPF attracts a lot of attention and resources from the scientific and medical community [4].

Despite COPD and IPF having very different functional characteristics (obstructive in COPD versus restrictive in IPF), they both involve an increased decline in lung function over time, measured by FEV_1_ in COPD [8,29,30] and FVC in IPF [31,32]. Interestingly, data from recent trials conducted in patients with severe COPD [33,34] or IPF [31,32] suggest that patients with COPD have much worse spirometric impairment at baseline, with FEV_1_ mostly <50% predicted, whereas in IPF FVC at baseline is mostly around 70–80% predicted [31,32]. Nevertheless, patients with IPF have a much worse prognosis and faster decline in lung volumes than those with COPD, suggesting very different pathophysiology. For example, in trials conducted in subjects with similar characteristics (65 years of age, smokers or ex-smokers), the decline in FEV_1_ is usually around 50–60 mL/year in patients with COPD [29], compared to an FVC decline of 150–300 mL in patients with IPF [31,32].

Obviously, FEV_1_ changes in COPD and are not comparable to FVC changes in IPF, especially in relation to outcomes such as mortality, given the very different disease processes: COPD is associated with alveolar and bronchiolar destruction and remodeling, with reduced elasticity, lung volumes, and lung compliance leading to hyperinflation. In contrast, IPF is associated with increased collagen deposition in the interstitium, increased elasticity, and decreased lung compliance and lung volumes. Thus, differences in mortality may not be explained by differences in lung function but rather by different pathophysiological mechanisms.

The effect of therapy also seems markedly different in the two diseases. A meta-analysis of randomized clinical trials (RCTs) of 1–4 years duration has shown that in patients with moderate-to-severe COPD, the average decline with active treatments (inhaled corticosteroids, long-acting β_2_-agonists, long-acting muscarinic antagonists, and their combinations) was in the range of 5–10 mL/year, which was 10–15% of the 50–60 mL/year decline in the placebo group. Two pivotal trials have been conducted in the last 10 years to examine the impact of treatment in patients with IPF, one on nintedanib and one on pirfenidone, [31,32] both of which were powered on FVC decline as the primary outcome [31,32]. In the ASCEND trial, King et al. [31] reported that the linear slope FVC decline at week 52 was 164 mL in the pirfenidone group and 280 mL in the placebo group (absolute difference, 116 mL; relative difference, 41.5%; *p* < 0.001). While we are discussing diseases with very different pathophysiologies, it is interesting that the effect of active treatment appears to be much larger in IPF than COPD, suggesting that these new agents might have the potential to revere remodeling and improve survival. However, this is just speculative, as no RCT has been powered to assess the effect of these treatments on survival, although a trend to reduced mortality has been reported in both the ASCEND trial [31] and a metanalysis of all phase 3 clinical trials of pirfenidone [35].

Another interesting difference between the two diseases is that most recent COPD phase III RCTs have been powered on exacerbations [33,34] with lung function as a secondary outcome, whereas IPF RCTs are typically powered on lung function (FVC) decline with exacerbations as a secondary outcome, although mortality is often included as a secondary outcome in both [31,32,33,34]. Surprisingly, even if the relative effect of treatment on lung function in COPD is modest compared to IPF, the effect of treatment on mortality is much greater in COPD [33,34] than in IPF [31,32]. Similarities and differences between COPD and IPF should be further explored.

Albeit with a very different time-course, these three chronic respiratory diseases (COPD, ILAs, IPF) manifest clinically with a very similar pattern, i.e., progressive dyspnea that is often underestimated because it is considered a physiologic effect of aging and smoking, with episodes of exacerbations of dyspnea and, in different proportions, other respiratory symptoms (cough, sputum, and wheezing). They are almost invariably diagnosed in elderly smokers and are associated with an increased risk of lung cancer, multimorbidity, and mortality, with mortality being higher and sooner in IPF [12,36,37].

Aging and smoking are also major risk factors for other chronic diseases that often develop together with respiratory diseases, particularly cardiovascular and metabolic diseases, contributing to the worldwide epidemic of chronic multimorbidity [36]. These chronic diseases also often manifest with progressive dyspnea and episodes of exacerbations of dyspnea [36]. Aging and smoking are also associated with an increased risk of lung cancer, which is also associated with these chronic respiratory diseases [37].

## 5. Aging-Related Pathways in COPD and IPF

Aging is associated with impairment in the homeostasis of organs and systems, including the lungs [38]. Aging is a complex process including cellular senescence, telomere shortening, epigenetic changes, genomic instability, mitochondrial dysfunction, and altered cellular interactions

Cellular senescence is characterized by permanent cell cycle arrest and the formation of the complex secretory phenotype known as SASP (senescence-associated secretory phenotype) [39]. Cellular senescence is induced not only by telomere shortening but also by different types of cellular stress, including oxidative stress and tobacco smoking. In SASP, senescent cells affect the microenvironment by acting on gene expression of different cytokines, proteases, and growth factors. Senescent cells thus remain metabolically active and are capable of altering their microenvironment, inhibiting tissue repair and acting as a source of chronic inflammation. Moreover, through their mediators, they may induce further senescence to the cell itself (autocrine) and to surrounding cells (paracrine), amplifying and spreading cellular senescence. Autophagy is a physiological process of lysosomal self-degradation that maintains the homeostatic balance between synthesis, degradation, and recycling of cellular proteins that diminishes with aging [40]. Cellular senescence and autophagy have been widely investigated in IPF and COPD [41], but while increased cellular senescence has been found in both diseases, reduced autophagy has only been found in IPF; data in COPD are still controversial.

In patients with IPF, re-epithelialization of remodeled air spaces with bronchial epithelial cells is a prominent pathological finding of the disease; recent studies suggest that such re-epithelialization may be influenced by the acceleration of cellular senescence. Interestingly, in vitro models using cultured primary human bronchial epithelial cells (HBEC) by Minigawa et al. [42] showed that transforming growth factor β (TGF-β), a profibrotic mediator, induces cellular senescence and increases expression of Sirtuin 6 (SIRT6), a class III histone deacetylase *(**HDAC**)* expressed in epithelial cells in an IPF lung. Although the precise antisenescence functions of SIRT6 in IPF need to be clarified, increased expression of SIRT6 might be an insufficient compensatory mechanism against stress-induced cellular senescence by proapoptotic stimuli such as TGF-β, resulting in accelerated cellular senescence in this disease [42].

Persistent senescence has two deleterious consequences: (1) due to cell-cycle arrest, senescence may cause stem/progenitor cell renewal dysfunction, with loss of repair capability, and (2) SAPS may induce proinflammatory and remodeling cytokines, such as TGF-β, promoting lung fibrosis [43]. Moreover, defective autophagic activity mediated by mTORC1 (mammalian target of rapamycin complex 1) has been found in fibroblasts with increased resistance to apoptosis in IPF lung fibroblasts [44], a potential mechanism of further amplification of lung fibrosis.

In patients with COPD, senescence and autophagy of alveolar cells and fibroblasts may be responsible for structural changes of (1) proximal airways, with chronic bronchitis and mucus production, and peripheral airways, with airway inflammation and remodeling, with the disappearance of small conducting airways [14]; and (2) lung parenchyma, with destruction and loss of alveolar attachments leading to emphysema [14]. Cellular senescence has been found in lung epithelial and endothelial cells and fibroblasts from patients with COPD [45,46,47].

Sirtuins, in particular SIRT1, an anti-inflammatory and antiaging protein, are decreased in peripheral lungs of smokers and patients with COPD, particularly in alveolar macrophages and alveolar epithelium, as a result of oxidative stress and tobacco smoking [48]. The decreased expression of SIRT1 results in an increase in proinflammatory cytokines through nuclear factor kappa-light-chain-enhancer of activated B cells (NF-kB) activation and acceleration of cellular senescence through a decrease of anti-senescence activity, mediated by FOX3 [48]. In addition to the effects in IPF, SIRT6 may be implicated in the development of COPD through increasing cell senescence. SIRT6 also regulates longevity by modulating the insulin-like growth factor (IGF)-1 pathway [49]. Indeed, SIRT6 antagonizes cell senescence through the attenuation of IGF-1, and the observation that SIRT6 expression is reduced in the lungs of smokers and COPD patients support this hypothesis. Interestingly, decreased expression of SIRT6 has been associated with decreased lung function (FEV_1_) in patients with COPD [50].

It is intriguing that while cellular senescence seems to be implicated in the pathogenesis of COPD, in terms of impaired cell repopulation and tissue destruction, in the IPF lung, increased cellular senescence seems to be involved in cell proliferation and tissue remodeling with the progression of fibrosis [47,51].

*Telomere dysfunction*. In humans, telomeres consist of a repeating sequence of TAAGGG hexanucleotide located at the ends of chromosomes. They have an important role in maintaining chromosome integrity and cell proliferation and progressively shorten with age. Telomerase is the specialized polymerase that synthesizes new telomere repeats [52,53,54]. It has two core components: TERT, telomerase reverse transcriptase, and TR, telomerase ribonucleic acid (RNA), which provides the template for telomere repeat addition [54,55].

Telomere shortening, mutation, and dysfunction leading to abnormal deoxyribonucleic acid (DNA) repair and genomic instability may heavily influence human lung aging. Such changes have been found both in IPF and COPD, even if its causative role in COPD is debated [56].

Telomere shortening may increase the risk of IPF, negatively influence disease progression, and contribute to a worse prognosis after lung transplantation. In patients with both familial and sporadic IPF, diminished telomerase activity and prematurely shortened telomere length (TL) have been found in blood leukocytes regardless of the presence of genetic mutations in TERT (Telomerase Reverse Transcriptase) and/or in TERC (telomerase RNA component) [57]. In addition, shorter TLs have been described in lung alveolar type 2 (AT2) cells compared to controls [58]; interestingly, shorter TLs have also been described in AT2 cells in fibrotic areas compared to the non-fibrotic area in the same biopsy. Furthermore, short lung TL is also associated with decreased survival [59]. Together, these findings indicate that telomere-related pathology plays a crucial role in both familial and sporadic IPF. A study by Duckworth and co-workers using a mendelian approach further supports a causal link between TL and IPF development [56].

Patients with COPD present shorter leucocyte telomeres than smokers without COPD and healthy subjects [60,61]. Further, compared to smoking controls, patients with COPD exhibit an accelerated telomere shortening that is associated with worse alveolar gas exchange, worse lung hyperinflation, and the presence of extrapulmonary affections [62]. Moreover, in patients with COPD, shorter telomeres are associated with an increased risk of cancer and all-cause mortality [62,63]. However, when mendelian randomization was used to investigate causality between TL and incidence of COPD in a large cohort of 11,413 patients with COPD and more than 400,000 controls, there was no evidence that telomere shortening may cause COPD [56]. Further studies are required to unravel the relationship between telomere shortening and the development of COPD.

A schematic representation of potential common mechanisms related to aging and senescence operating in IPF and COPD is in Figure 2.

## 6. Genetic Susceptibility to IPF and COPD

Both IPF and COPD are complex genetic diseases, featuring interactions between genetic, environmental, and occupational factors, in particular tobacco smoking, subclinical (viral) infections, gastroesophageal reflux disease (GERD), and familial disease. Genome-wide search studies have been performed in patients with IPF or COPD, revealing complex and different genetic variants conferring susceptibility to the two diseases.

Genome-wide association studies (GWAS) [64,65,66,67] have identified 17 chromosomal regions associated with increased susceptibility to IPF. To date, the most strongly associated genetic variant associated with an increased risk of IPF is located in the promoter region of the mucin 5B (MUC5B) gene [68]. MUC5B is a mucin with a crucial role in maintaining effective mucociliary clearance in the airways. The common rs35705950 polymorphism leads to increased accumulation of MUC5B, with impairment of mucociliary clearance resulting in chronic exposure of distal airways to inhaled particles and pathogens. In genetically susceptible individuals, this may initiate cell injury and promote the fibroproliferative response. Paradoxically, patients with IPF carrying the common mutant rs35705950 promoter polymorphism had longer survival, a finding that remains difficult to explain [69]. Given this, polymorphisms in the MUC5B gene continue to be a focus of ongoing studies.

Genome-wide search studies also identified different rare genetic variants in genes involved in the maintenance of TL (TRGs, telomere-related genes) associated with an increased risk of IPF development. Indeed, mutations have been found in the TERC (Telomerase RNA component) gene, in the poly(A)specific ribonuclease (PARN) gene that is involved in telomere maturation, and RTL (regulator of telomere elongation helicase)1, a gene involved in DNA helicase activity. Mutations in TRGs genes have mostly been reported in familial IPF, although they have also been recently found in sporadic IPF, supporting the role of aging in the development of IPF. The telomerase complex genes have been extensively reviewed by Stock and Renzoni [70]. In familial IPF, different mutations have been described in the surfactant protein genes that are involved in maintaining alveolar structure and host defense.

The largest GWAS of IPF susceptibility performed to date combined all previous IPF genome-wide association studies and confirmed 11 of the 17 previously associated regions but identified three novel regions. These three regions implicate the genes DEPTOR, KIF1, and MAD1L1 [71], all of which are involved in the pathogenesis of IPF. Indeed, DEPTOR variants, by interacting with the mTORC1 and mTORC2 protein complex, may stimulate collagen synthesis [72], while MAD1L1 may increase IPF susceptibility by reducing telomerase activity [73].

Genetic factors also play an important role in COPD susceptibility. Although severe hereditary deficiency of α1-antitrypsin (A1AD) has been established to cause emphysema, A1AD accounts for only approximately 1% of COPD cases. Several genome-wide association studies have been successful at detecting more than 50 loci predicting disease or lung function impairment risk; pathways that play a role in COPD development include the response to oxidative stress, the protease–antiprotease imbalance, inflammation, and senescence. Currently, the most well-known candidate genes for COPD, which have been replicated in multiple populations, are CHRNA3 and CHRNA5 (cholinergic nicotine receptor alpha 3/5), involved in nicotine addiction, FAM13A (family with sequence similarity 13, member A), involved in the protection of the airway epithelium against environmental exposure [74], IREB2 (iron regulatory binding protein 2), involved in neutrophilic inflammation, HHIP (Hedgehog-interacting protein), involved in the development and repair of lung tissue, and AGER (advanced glycosylation end product-specific receptor), involved in the pathogenesis of COPD [75]. In another GWAS [76] that combined data from the International COPD Genetics Consortium (ICGC) with additional subjects from the UK Biobank, the authors identified 82 genome-wide significant loci for COPD, of which 47 were previously identified in GWAS of COPD or population-based lung function. Interestingly, they reported five shared genome segments with pulmonary fibrosis, FAM13A, DSP, and 17q21 already known [77], and two new segments, including loci near ZKSCAN1 and STN1 (formerly known as OBFC1). Shared variants between COPD and IPF all had an opposite effect (i.e., increasing risk for COPD, but protective for IPF).

In conclusion, genetic factors play an important role both in IPF and COPD susceptibility, but given that COPD and IPF are so different in their pathogenesis, pathology, natural history, and clinical manifestations, the genetic factors differ and carry a different weight in each of them.

## 7. Severe Exacerbations of COPD, IPF, and Other Interstitial Lung Diseases

Up to 3–4% of cases of COPD and IPF are associated with episodes of severe acute exacerbations, defined as acute deteriorations of respiratory symptoms that lead to the need for additional therapy and/or hospitalization and, in the most severe cases, to the development of respiratory failure that may require ventilatory support. They are usually more severe and often fatal in IPF [27].

In both diseases, acute exacerbations are heterogeneous and complex events, whose underlying mechanisms are still poorly understood and which may be triggered by infections (particularly in COPD), environmental factors, or GERD, but may also be idiopathic, particularly in IPF. A key feature of acute exacerbations of COPD (ECOPD) is an aberrant acute inflammatory burst on top of chronic inflammation, which occurs mainly in the airways. Acute exacerbations of IPF (AE-IPF) are characterized by diffuse alveolar damage, mainly affecting the alveoli and the interstitium. The resulting functional respiratory changes and outcomes differ according to the severity of the exacerbation and the underlying disease (much worse in COPD). However, despite IPF being the most common and severe form of interstitial lung disease (ILD), ILD represents a group of heterogeneous clinical conditions of both idiopathic and secondary natures that can develop an acute exacerbation in the course of the disease. Similar to patients with ILD or COPD, those with CPFE are at risk of an acute exacerbation [19], although limited data are available on the characteristics, relapses, therapeutic management, and prognosis of CPFE exacerbations.

## 8. Severe Exacerbations of COPD

About 30% of patients with COPD have at least one ECOPD per year [12,13]. These events have a significant impact on patients’ quality of life and disease course, as they are associated with accelerated lung function decline and increase both health care resource use and mortality. They are severe in 3–4% of cases [13,78,79].

The term ECOPD includes an umbrella of different conditions characterized by diverging underlying mechanisms and responses to treatments [13,78,79]. Despite COPD being considered a chronic airway inflammatory disease predominantly driven by neutrophils and CD8 +T lymphocytes, 20–40% of patients also exhibit eosinophilic airway inflammation, which predisposes them to a better response to corticosteroids, but also to increased ECOPD risk [80,81]. The main triggers of ECOPD are infections (viral and bacterial), but they can also be associated with non-infectious stimuli, including eosinophilic inflammation, air pollution, pauci-inflammatory events, and even worsening of cardiovascular disease [82]. Studies that divided exacerbations into subgroups based on the endotype (bacterial, eosinophilic, and viral/other, pauci-inflammatory) described bacterial ECOPDs as more symptomatic and severe than eosinophilic ECOPDs [82].

The GOLD strategy document defines ECOPDs as severe when they result in either hospitalization or a visit to the emergency department and/or are associated with respiratory failure [83]. Although mechanisms underlying susceptibility to the development of severe ECOPD are not known, some biological factors may be involved. Impaired interferon gamma (IFN-γ) induction by bronchial epithelial cells in response to viral infections appears to be a common feature in severe COPD and may contribute to susceptibility to viral infections and severe ECOPDs [84]. Experimental data show that increased susceptibility to influenza infection in COPD occurs through reduced induction of protein kinase R, an IFN-stimulated gene that is directly involved in limiting viral replications and in inducing IFN-γ through the binding of the recognition receptor, retinoic-acid inducible gene (RIG)I, to the adaptor mitochondrial signaling (MAVS) on the mitochondria. In addition, in bronchial epithelial cells from subjects with COPD, enhanced levels of microRNA (miR)-132 reduce transcriptional coactivator p300 expression, contributing to impaired formation of antiviral stress granules and IFN-γ enhanceosome [85]. Further, alveolar macrophages obtained by bronchoalveolar lavage from patients with COPD who have an ECOPD phenotype showed impaired cytokine induction (TNF-γ, interleukin [IL] 8) after exposure to bacteria, compared with macrophages from patients with COPD free from ECOPD [86].

Increased severity and frequency of ECOPD could also be due to a relative immune dysfunction, as indicated by an increased number of functionally suppressive circulating cells and effector T-cell dysfunction [87]. Indeed, peripheral lymphocytes obtained from patients with ECOPD exhibit impaired responsiveness to bacterial antigens compared to those with stable COPD and healthy controls [88]. Effector T-cell dysfunction in COPD has been related to the accumulation of functionally suppressive Tregs, a subset of CD4^+^ T cells which regulate immune response and establish peripheral tolerance, but which also play a decisive role in controlling effector T-cell function through the secretion of inhibitory cytokines such as TGF-β, IL-35, and IL-10 [89]. However, in patients with COPD, effector T-cells are not only suppressed by Tregs but also exhibit an exhausted phenotype with upregulation of programmed death-1 (PD-1) protein, a negative co-stimulatory molecule that impairs immunity by inducing apoptosis, increasing IL-10 production, preventing T-cell proliferation, and reducing T-cell reactivity [90]. PD-1, by binding its ligands PD-L1 and PD-L2, which are expressed in a wide variety of cells, exerts its inhibiting effect on CD8^+^ functions. However, in a murine model, functionally exhausted T-cells were defined by co-expression of PD-1 and T-cell immunoglobulin and mucin domain (TIM)-3, which was not detected in T-cells isolated from lung tissue of patients with COPD. Indeed, McKendry and colleagues showed that in COPD lung tissue, the defective response to viruses was associated with impaired cytotoxicity of CD8 T-cells that overexpressed PD-1 but did not express detectable surface TIM-3. They also found a reduction in infection-induced expression of PD-L1 on COPD macrophages with a concomitant increase in IFN-γ release from CD8^+^ cells [91]. These data indicate that viral ECOPD may be sustained by CD8^+^ T cell populations, which do not fulfill the classic criteria of T-cell exhaustion, exhibiting impaired viral activity, coupled with an inability to downregulate cytokine secretion. The consequence of this complex immune process is the coexistence of exaggerated inflammatory response and impaired host defense, which results in bronchial damage and viral susceptibility. During ECOPD, the infection elicits recruitment of neutrophils, macrophages, T-cells, and dendritic cells at the site of infection, with an airway inflammation response that is more relevant in COPD than healthy controls [78]. Further, in ECOPD, the inflammation can overflow from the airways towards the circulation, as demonstrated by the finding of increased concentrations of circulating systemic inflammatory biomarkers such as C-reactive protein, fibrinogen, and IL-6 [92]. The elevation of systemic inflammation may be linked to an increased risk of subsequent cardiovascular events within the first 30 days after ECOPD in hospitalized patients with COPD and comorbid cardiovascular disease, as reported by SUMMIT (Study to Understand Mortality and Morbidity) [93].

The pathophysiologic cascade triggered by severe ECOPD may result in profound alterations in respiratory mechanics and gas exchange, contributing to the high mortality in patients with ECOPD requiring hospitalization [94]. Acute airway narrowing and the consequent increase in airway resistance associated with breathing pattern modification (i.e., shallow breathing pattern) lead to a significant mechanical respiratory change. Airways narrowing and reduction in expiratory time worsen expiratory flow limitation and create regions in the lungs that cannot properly empty and return to their normal resting volume; alveolar pressure at the end of expiration, therefore, remains positive, resulting in auto- or intrinsic dynamic, positive end-expiratory pressure (PEEP_i_, dyn) [95]. Thus, during ECOPD, dynamic hyperinflation increases end-expiratory lung volume and shifts tidal volume to an area of the pressure–volume curve where the lung is less distensible, resulting in an increased elastic load that, together with the increase in the resistive load, contributes to the imbalance between respiratory muscle effort and load. The consequences of volume overload during ECOPD are the flattening and the shortening of the diaphragm, resulting in mechanical disadvantage and impairment of its contractile capacity, ranging from muscle dysfunction to paralysis [96]. Diaphragm dysfunction seems more relevant in patients with COPD who have lungs with extremely low elastance value, probably because this feature constitutes an elastic substrate of the lung, favoring acute volume overload when PEEP_i_, dyn develops [96]. Moreover, dynamic hyperinflation may also result in cardiac function impairment due to biventricular end-diastolic volume reduction and consequent reduction in pre-load and cardiac output, which could decrease the perfusion of respiratory muscles and further promote diaphragm weakness [97]. The complex interactions between these factors in the setting of severe ventilation–perfusion abnormalities (i.e., high physiological dead space) result in carbon dioxide retention and arterial oxygen desaturation.

Treatment of severe ECOPD is based on the use of inhaled bronchodilators, systemic corticosteroids, antibiotics, and, for hypoxemic patients, supplemental oxygen [98]. In ECOPD patients with acute respiratory failure leading to hypercapnia and respiratory acidosis (pH ≤ 7.35), noninvasive ventilation (NIV) is recommended to prevent intubation and invasive mechanical ventilation (MV) [99]. Two studies compared NIV and MV in patients with ECOPD who were considered to require endotracheal intubation (mean pH < 7.2). Survival was similar in both groups, but patients successfully treated with NIV showed a shorter duration of intensive care unit (ICU) and hospital stay, fever episodes of ventilator-associated pneumonia, and reduced need for oxygen supplementation [99,100]. Therefore, recent European Respiratory Society/American Thoracic Society guidelines recommend that a trial of bi-level NIV may also be attempted as an alternative to first-line endotracheal intubation unless patients show signs of immediate respiratory deterioration [99].

In conclusion, ECOPDs represent an important clinical manifestation of COPD that increases in frequency and severity with the duration and severity of the underlying disease, accelerates its progression, and increases mortality. When severe, ECOPDs require hospitalization and/or admission to ICU with assisted ventilation. Complex immunologic mechanisms may underly the susceptibility not only to severe ECOPD but also the complexity of underlying COPD that is almost invariably associated with chronic multimorbidity. This may explain why ECOPD and COPD itself present in a range of very different clinical and pathophysiological patterns [13,78].

## 9. Severe Exacerbations of ILD

Acute exacerbations of interstitial lung diseases (AE-ILDs), among which AE-IPF is the best known, can occur at any time along the course of the disease and constitute a serious life-threatening event in most cases. AE-ILD triggers can be similar to those of severe ECOPD, such as infections and air pollution, but AE-ILDs can also be triggered by causes including surgical lung biopsy, bronchoscopy (bronchiolar-alveolar lavage, cryobiopsy), GERD, and some medications (e.g*.,* methotrexate or tocilizumab) [101]. However, in many cases, no external triggers are identified; such AE-ILDs are termed *’idiopathic’*. Growing evidence suggests a key role of occult viral infections in the development of AE-IPF. Indeed, different viruses have been detected using molecular techniques in the course of AE-IPF, such as human herpes virus (HHV), torque teno virus (TTV), respiratory syncytial virus, influenza A, and cytomegalovirus [102]. Furthermore, studies that investigated the role of the respiratory microbiome in IPF have shown a marked change during AE-IPF compared to stable disease, but it is unclear whether this alteration reflects an active infection, a consequence of aspiration, or occurs as a result of diffuse alveolar damage [103].

A definition of AE-IPF was proposed by the International Working Group for AE-IPF as an acute and clinically significant respiratory deterioration characterized by radiological evidence of new widespread alveolar abnormality, typically less than 1 month in duration [104]. Cardiac failure or fluid overload should be excluded. AE-ILDs are associated with the development of severe acute respiratory failure with similar features to acute respiratory distress syndrome (ARDS), histologically and radiologically characterized by diffuse alveolar damage overlapping an underlying interstitial disease, leading to a poor prognosis and high mortality rate [105]. The overall risk of acute exacerbations is particularly high in IPF, with an annual incidence ranging from 5% to 19%. [106]. In other fibrosing ILDs, especially chronic hypersensitivity pneumonitis, asbestosis, fibrosing non-specific interstitial pneumonia (NSIP), and connective tissue disease (CTD-ILD), the annual risk of acute exacerbations varies widely among studies and is associated with radiological and histological patterns of usual interstitial pneumonia (UIP) [101].

Although the physiopathology of AE-ILD has not yet been fully understood, different triggers could stimulate overexpression of alveolar macrophage M1 proinflammatory cytokines, such as IL-8 and C-X-C motif ligand 1 (CXCL1), that promote alveolar neutrophil recruitment and subsequent diffuse alveolar damage. Furthermore, simultaneous activation of M2 macrophage pathways results in overexpression of cytokines with a profibrotic role, leading to the progression of pulmonary fibrosis [105]. Evidence from animal models suggest that Tregs could play an important regulatory role in acute lung injury and AE-IPF. Moyé and collaborators, in a mice model of IPF, studied the potential role of Tregs in the modulation of infection-driven AE-IPF. They observed that Treg depletion worsened the severity of AE-IPF, whereas IL-2 complex-induced expansion of Tregs leads to an attenuation, suggesting that Tregs could act as a regulator of the immune response during AE-IPF through downregulation of lung inflammatory cytokines, including TNF-α, IL-6, and TGF-β [107]. Finally, AE-ILDs result in acute respiratory failure and severe hypoxemia due to an increased intrapulmonary shunt, with consequent strong activation of the respiratory drive. Further, the sudden deterioration of lung mechanics causes a further increase in inspiratory efforts to maintain adequate minute volume ventilation, resulting in excessive pleural pressure swings, promoting self-inflicted lung injury (SILI) and progression of alveolar damage [108].

The optimal treatment strategy of AE-ILD is not yet known as there are no randomized trials establishing the efficacy of one drug over another or placebo. Limited data from retrospective studies suggest a benefit from the use of corticosteroids in most AE-ILD, although post-AE-ILD mortality remains high [24,104]. However, systemic corticosteroids are ineffective, if not harmful, in AE-IPF [109] and in systemic sclerosis-associated ILD (SSc-ILD), especially in the latter, given the risk of associated SSc renal crisis, a potentially life-threatening complication that affects 2–15% of cases, particularly in patients with early, rapidly progressive, and diffuse cutaneous SSc [110].

Immunosuppressive agents (e.g., cyclophosphamide, tacrolimus, azathioprine, and cyclosporin A) have been used in clinical practice, based on small retrospective trials or anecdotal reports, with no robust evidence to support their role in AE-ILD [111]. Immunosuppression may have a role in AE-ILDs such as hypersensitivity pneumonitis [112] and CTD-ILD exacerbations [113].

Empiric broad-spectrum antibiotics are often used in patients with AE-ILD due to the difficulty in carrying out specific diagnostic tests (such as bronchiolar–alveolar lavage) that exclude the role of bacterial infections in AE-ILDs. In patients with acute respiratory failure presenting with severe hypoxemia and excessive inspiratory effort, a trial of NIV could be offered to support spontaneous breathing, improve gas exchanges and reduce work of breathing—although in retrospective trials, mortality remains high in patients with AE-ILD undergoing NIV (between 45 and 75%). However, early application of NIV could result in a survival advantage in patients with less severe general conditions [105]. High-flow oxygen could be proposed in patients who fail conventional oxygen therapy, although data on the benefit of this method are lacking [114].

Endotracheal intubation and MV are associated with high mortality and progression of pulmonary fibrosis, and they are therefore only offered in selected patients and as a bridge to lung transplantation. In patients undergoing MV, parameters suggested for protective ventilation in ARDS (i.e., tidal volume less 6 mL/kg, plateau pressure <30 cmH_2_O, driving pressure <15 cmH_2_O) must be carefully observed, although the extreme reduction in lung compliance makes it difficult to keep lung stress below harmful levels. However, in patients with AE-ILD with an overlapping UIP pattern, the different mechanical behavior of the lung compared to ARDS makes the fibrotic lung at risk of ventilator-induced lung injury (VILI), especially when high levels of PEEP are used [108]. Indeed, retrospective data showed an association between higher PEEP levels and mortality in patients with AE-ILD undergoing MV [115]. Therefore, in fibrotic lungs with superimposed diffuse alveolar damage, the adoption of a *’lung resting strategy*’ is suggested, with PEEP titrated to the minimal values necessary to achieve a satisfactory level of oxygenation (i.e., SpO_2_ above 88–90%).

In conclusion, AE-ILD are complex and crucial events in the natural history of the various conditions and are associated with a poor prognosis due to a lack of optimal, evidence-based management. Even the role of MV in severe respiratory failure due to AE-ILD is still debated, especially in patients with IPF, given the excessively high mortality.

## 10. Exacerbations of CPFE

Patients with CPFE may experience exacerbations that can be attributed to the emphysema component (COPD-type) or the fibrosis component of the lung (IPF-type) [116]. COPD types of AE-CPFE are defined in the same way as severe exacerbations of COPD (i.e., acute worsening of respiratory symptoms that result in hospitalization) and show clinical and physiological features comparable to ECOPD: airflow obstruction, increased airway resistance, and dynamic hyperinflation [20]. IPF types of AE-CPFE are defined according to the criteria established by the International Working Group for AE-IPF and are associated with higher morbidity and need for ICU management compared to the COPD type [104,116]. Initial findings have shown that the baseline pattern on the CT-scan (predominance of emphysema, or fibrosis, or both) strongly correlates with the type of AE-CPFE [116]. The exact annual risk of AE-CPFE is unknown but appears to be less frequent than AE-IPF. A retrospective study that compared the clinical outcome of patients with AE-CPFE with that of patients with AE-IPF showed a significantly better prognosis and a higher survival rate in AE-CPFE [117]. The treatment and management of patients with AE-CPFE reflect the type of underlying exacerbation (COPD-type or IPF-type).

## 11. Conclusions and Perspective

IPF and COPD are two distinct smoking-related chronic pulmonary diseases characterized by different lung pathologies and functional impairments, although both are associated with acute exacerbations that impact patients’ quality of life and survival. In both, aging has a key role in disease development; however, genetic background, such as epigenetic influences, may be responsible for the activation of different molecular pathways.

To understand why some older smokers develop IPF, COPD, or the combined phenotype, different approaches are necessary, including genetic studies on susceptibility (e.g., genome-wide search studies), experimental studies to identify different molecular pathways, and clinical studies to better characterize (e.g., through lung function and imaging) the cross-talk between pulmonary fibrosis, bronchiolitis, and emphysematous abnormalities.

## Figures and Tables

**Figure 1 ijms-22-09292-f001:**
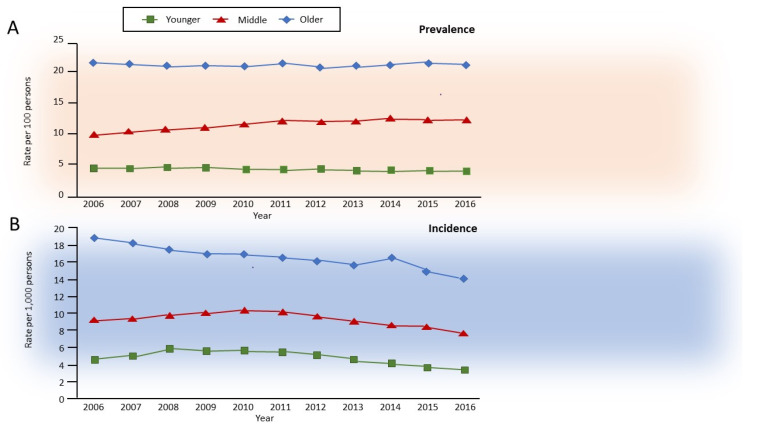
Annual age-specific standardized (**A**) prevalence and (**B**) incidence of chronic obstructive pulmonary disease in Ontario, Canada, 2006–2016. Rates are sex-standardized to the 2016 Ontario population, with age groups defined as younger, 35–54 years old; middle, 55–64 years old; and older, 65 years and older. Reprinted with permission of the American Thoracic Society. Copyright © 2021 American Thoracic Society. All rights reserved. Gershon AS, McGihon RE, Luo J, Blazer AJ, Kendzerska T, To T, Aaron SD. Trends in Chronic Obstructive Pulmonary Disease Prevalence, Incidence, and Health Services Use in Younger Adults in Ontario, Canada, 2006–2016. Am J Respir Crit Care Med 2021: 203(9): 1196–1199. The American Journal of Respiratory and Critical Care Medicine is an official journal of the American Thoracic Society.

**Figure 2 ijms-22-09292-f002:**
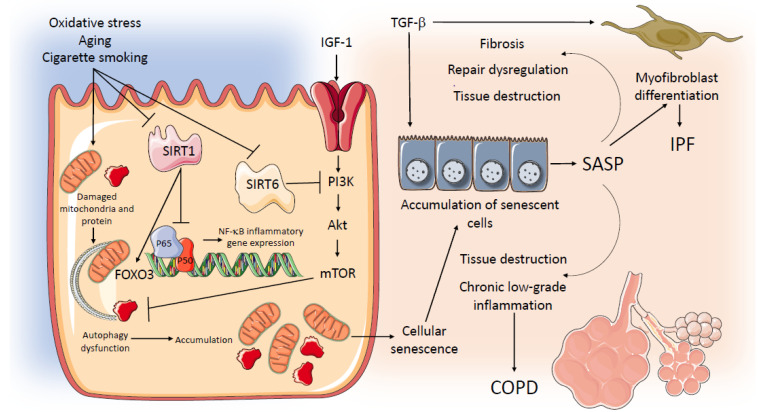
Potential common mechanisms related to aging and senescence operating in IPF and COPD. AKT: serine/threonine kinase; COPD: Chronic Obstructive Pulmonary Disease; FOXO 3: Forkhead Box O3; HBEC: human bronchial epithelial cells; IPF: Idiopathic Pulmonary Fibrosis; IGF-1: Insulin Growth factor 1; mTOR: Mechanistic Target of Rapamycin; NF-kB: nuclear factor kappa-light-chain-enhancer of activated B cells; PI3K: phosphoinositide 3-kinase; SASP: senescence-associated secretory phenotype; SIRT 1: Sirtuin-1; SIRT 6: Sirtuin-6; TGFβ: Transforming Growth Factor β.

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
