# Peer review of "COPD, Pulmonary Fibrosis and ILAs in Aging Smokers: The Paradox of Striking Different Responses to the Major Risk Factors"

_ijms, 2021, doi:10.3390/ijms22179292_

Round 1
Reviewer 1 Report
Although COPD, IPF and CPFE are different lung diseases, they share risk factors and specific associations such as history of smoking, male sex and certain occupational exposures. The authors have undertaken a review of these conditions focusing on the role of ageing and smoking in disease pathophysiology.
Whilst some sections of the article are clear and understandable, some sections are confusing and require major revision.
Comments:
- The second part of the title of the review article is ambiguous. I suggest this is changed “….the paradox of different pathophysiological responses to shared risk factors” or something similar to this
- Interstitial lung abnormalities are more prevalent in older individuals and can occur in both smokers and non-smokers. They can progress to idiopathic pulmonary fibrosis, particularly in those with MUC5B polymorphism associated with IPF. The syndrome of CPFE is distinct to ILAs. In CPFE, several different types of fibrosis have been described including UIP and NSIP. ILAs are not discussed elsewhere in the manuscript and I am not clear why it is included. Is it required?
- Section 3: idiopathic pulmonary fibrosis – the comparisons here between COPD and IPF are very confusing. I suggest keeping this focused on IPF. It may be better to discuss the lung function abnormalities in a separate section focusing on pathophysiology of these conditions. Alternatively, this could be discussed in each of the sections about COPD, CPE and IPF. In CPFE, need to discuss the combination of emphysema and fibrosis results in relatively normal spirometry (FEV1 and FVC) whilst the gas diffusion co-efficient is significantly reduced. This is associated with higher risk of pulmonary hypertension and poor mortality. In section 3, FEV1 data for COPD and FVC data for IPF with regards to mortality and outcomes of clinical trials are not comparable. IPF and COPD are different disease processes – one resulting in increase in collagen in the interstitium, decreasing lung compliance and reduced lung volumes; whereas alveolar destruction, increased lung compliance and reduced elasticity occurs in COPD leading to hyperinflation, which may increase lung volumes. Differences in mortality are not explained by the differences in lung function parameters – it is better explained by different disease pathogenesis. In IPF, the clinical trials are not powered to detect a change in mortality; they are powered on change in FVC at 1 year. Please review comment about IPF clinical trials on page 4 line 145.
- Section 8: exacerbations IPF – this section has confused acute exacerbations of IPF and other ILDs. There is no evidence for the use of corticosteroids or immunosuppression in AE-IPF – see Farrand E et al Respirology 2020, 25(6):629-635. Immunosuppression may have a role in acute exacerbations of other ILDs such as hypersensitivity pneumonitis and CTD-ILD.
- The manuscript would be much improved with a conclusion or summarising section. This could also include areas for future research and development .
- Introduction page 1 line 22 and page 3 line 113 – please change to 0. 0005-0.002%.
- page 1 line 36 change to 5, 1.0, 1.6%
- page 1 line 37 – please review as it does not make sense “…roughly 5, 11, 23 % in the [9] (Figure 1) “
- Figures – please clarify if figure 1 has been reproduced with permission from reference 9 or has it been modified by the authors?
- There are several grammatical errors throughout the manuscript, please can these be corrected.
Author Response
GENERAL COMMENTS
We thank the Editors and the Reviewers for their constructive comments. In addition to re-write some sections in response to general comments of the reviewer, and the request to have the manuscript edited by a native English mothertongue scientific editor, we rewrote particularly Sections 1 and 3, and marked in bold italics specific responses.
REVIEWER 1 (R1) - COMMENT 1 (C1)- R1.C1.
Although COPD, IPF and CPFE are different lung diseases, they share risk factors and specific associations such as history of smoking, male sex and certain occupational exposures. The authors have undertaken a review of these conditions focusing on the role of ageing and smoking in disease pathophysiology.
Response to R1.C1. (RR1.C1.)
We thank the reviewer for her/his nice summary
R1.C2.
Whilst some sections of the article are clear and understandable, some sections are confusing and require major revision.
The second part of the title of the review article is ambiguous. I suggest this is changed “…. the paradox of different pathophysiological responses to shared risk factors” or something similar to this
Response to R1C2 (RR1.C2.)
We thank the reviewer for the suggestion. The title was changed accordingly and now reads, “COPD, PULMONARY FIBROSIS AND CPFE IN AGEING SMOKERS: the paradox of different pathophysiological response to similar risk factors”
R1.C3.
Interstitial lung abnormalities are more prevalent in older individuals and can occur in both smokers and non-smokers. They can progress to idiopathic pulmonary fibrosis, particularly in those with MUC5B polymorphism associated with IPF. The syndrome of CPFE is distinct to ILAs. In CPFE, several different types of fibrosis have been described including UIP and NSIP. ILAs are not discussed elsewhere in the manuscript and I am not clear why it is included. Is it required?
RR1.C3.
We thank the reviewer for providing clear descriptions and definitions of the different disease entities we have discussed in the article. We meticulously reported them in the manuscript to make it clearer to read. Regarding ILAs, we respectfully disagree with the reviewer, and we rather keep ILAs (Section 2, together with CPFE, line 110) in the manuscript, as we believe they represent a relevant disease entity associated with aging and smoking (lines 123-124) . Regarding CPFE, we explain the reason for including them in Section 2, clearly stating that CPFE represents a separate disease entity from ILAs.
R1.C4.
Section 3: idiopathic pulmonary fibrosis – the comparisons here between COPD and IPF are very confusing. I suggest keeping this focused on IPF. It may be better to discuss the lung function abnormalities in a separate section focusing on pathophysiology of these conditions. Alternatively, this could be discussed in each of the sections about COPD, CPFE and IPF.
RR1.C4
We thank the reviewer for this constructive comment and for the suggested revisions that we incorporated in the manuscript as described above. Briefly, we kept the focus on IPF in first part of Section 3, and discussed the specific features of the distinct diseases entities in the respective sections. In addition to re-write some sections in response to this general comments of the reviewer and to the request to have the manuscript edited by an native english mother-tongue scientific editor, we rewrote particularly Sections 1 and 3, and marked in bold italics only revisions in response to specific comments.
R1.C5
In CPFE, need to discuss the combination of emphysema and fibrosis results in relatively normal spirometry (FEV1 and FVC) whilst the gas diffusion co-efficient is significantly reduced. This is associated with higher risk of pulmonary hypertension and poor mortality.
RR1.C5
We thank the reviewer for this constructive comment and for the suggested revision that we incorporated in the manuscript (lines 135-140).
R1.C6.
In section 3, FEV1 data for COPD and FVC data for IPF with regards to mortality and outcomes of clinical trials are not comparable. IPF and COPD are different disease processes – one resulting in increase in collagen in the interstitium, decreasing lung compliance and reduced lung volumes; whereas alveolar destruction, increased lung compliance and reduced elasticity occurs in COPD leading to hyperinflation, which may increase lung volumes. Differences in mortality are not explained by the differences in lung function parameters – it is better explained by different disease pathogenesis.
RR1.C6.
We thank the reviewer for this specific and constructive comment, that we incorpored in the manuscript (lines 171-178)
R1.C7.
In IPF, the clinical trials are not powered to detect a change in mortality; they are powered on change in FVC at 1 year. Please review comment about IPF clinical trials on page 4 line 145.
RR1.C7.
As suggested we modified the comment on IPF clinical trials (lines 196-201)
R1.C8.
Section 8: exacerbations IPF – this section has confused acute exacerbations of IPF and other ILDs. There is no evidence for the use of corticosteroids or immunosuppression in AE-IPF – see Farrand E et al Respirology 2020, 25(6):629-635. Immunosuppression may have a role in acute exacerbations of other ILDs such as hypersensitivity pneumonitis and CTD-ILD
RR1.C8.
We thank the reviewer for this constructive comment. We revised the text to underline that AE-IPF are only one of the different AE-ILD, that they are not are not sensitive to corticosteroids or immunosuppressant agents (lines 584-587), and quoted the suggested reference Farrand E, Vittinghoff E, Ley B, Butte AJ, Collard HR. Corticosteroid use is not associated with improved outcomes in acute exacerbation of IPF. Respirology. 2020 Jun;25(6):629-635. doi: 10.1111/resp.13753. Epub 2019 Dec 17. PMID: 31846126) (reference 109).
R1C.9.
The manuscript would be much improved with a conclusion or summarising section. This could also include areas for future research and development.
RR1C.9.
As suggested by the reviewer we added a sentence of conclusions and/or a summarizing figure for each section of the manuscript
SPECIFIC COMMENTS
R1.SC1.
Introduction page 1 line 22 and page 3 line 113 (118 in the revised manuscript) – please change to 0.0005-0.002%
RR1.SC1.
Done
R1.SC2.
page 1 line 36 change to 0.5, 1.0, 1.6%
RR1.SC2.
Done
R1.SC3
page 1 line 37 – please review as it does not make sense “…roughly 5, 11, 23 % in the [9] (Figure 1)
RR1.SC3
In fact we deleted in the, as it was misleading, and replaced with the sentence in the same study
R1.SC4.
Figures – please clarify if figure 1 has been reproduced with permission from reference 9 or has it been modified by the authors?
RR1.SC4.
Figure was reproduced with permission from the same study
R1.SC5.
There are several grammatical errors throughout the manuscript, please can these be corrected
RR1.SC5.
The manuscript has been reviewed and heavily edited by an English native mother-tongue expert scientific editor (David Young, acknowledged in the manuscript).
Reviewer 2 Report
Please refer to the uploaded file for comments.

Author Response
The review by Beghe et al. indicated their focus as “The focus of this perspective is to discuss the paradox of the striking pathologic and pathophysiologic responses to the same risk factors, aging and smoking, COPD characterized by small airways fibrosis and remodeling and destruction of the lung parenchyma, and pulmonary fibrosis affecting almost exclusively the lung parenchyma and the interstitium. Albeit rare and almost unknown, we will also discuss some aspects of the mixed syndrome named combined pulmonary fibrosis and emphysema (CPFE)”
GENERAL COMMENTS
R2C1
Major concern: Overall, the review is a bit too lengthy and lacks focus. In the end, it is still unclear to me what is driving the difference in pulmonary phenotypes (airway vs. parenchymal diseases) of the old smoker population if that is the intention of the review. In the end, it seems that these two clinical entities are simply two different diseases that have common underlying factors-aging and smoking; please clarify
RR2C1
We agree with the comment of the reviewer, and in fact we included her/his sentence in the abstract (lines 31-32), introduction (lines 52-56), and in the general conclusions (line 654)
R2SC1
Does the review not have an abstract?
RRSC1
Abstract of 200 words included
R2SC2
I highly recommend that the authors add a conclusion or summary section at the end of the review. A summary in each section will also be beneficial for the readers to capture the objective.
RR2SC2
As suggested by both reviewers we added either a sentence or a summary figure at the end of each section, and at the end of the review
R2SC3
Line 224 “histone deacetylase (HADC).” I believe it should be HDAC, not HADC.
RR2SC3
The reviewer is correct: HADC should read HDAC, and has been corrected in the manuscript (line 251).